# Enhancing the Heterologous Expression of a Thermophilic Endoglucanase and Its Cost-Effective Production in *Pichia pastoris* Using Multiple Strategies

**DOI:** 10.3390/ijms241915017

**Published:** 2023-10-09

**Authors:** Wuling Dai, Haofan Dong, Zhaokun Zhang, Xin Wu, Tongtong Bao, Le Gao, Xiaoyi Chen

**Affiliations:** 1School of Biological Engineering, Dalian Polytechnic University, Dalian 116034, China; daiwl@tib.cas.cn; 2Tianjin Institute of Industrial Biotechnology, Chinese Academy of Sciences, National Technology Innovation Center for Synthetic Biology, Tianjin 300308, China; donghf@tib.cas.cn (H.D.); zhangzk@tib.cas.cn (Z.Z.); wuxin@tib.cas.cn (X.W.); baott@tib.cas.cn (T.B.)

**Keywords:** thermophilic endoglucanase, multifactorial regulation strategy, cost-effective production, single-cell protein

## Abstract

Although *Pichia pastoris* was successfully used for heterologous gene expression for more than twenty years, many factors influencing protein expression remain unclear. Here, we optimized the expression of a thermophilic endoglucanase from *Thermothielavioides terrestris* (TtCel45A) for cost-effective production in *Pichia pastoris*. To achieve this, we established a multifactorial regulation strategy that involved selecting a genome-editing system, utilizing neutral loci, incorporating multiple copies of the heterologous expression cassette, and optimizing high-density fermentation for the co-production of single-cell protein (SCP). Notably, even though all neutral sites were used, there was still a slight difference in the enzymatic activity of heterologously expressed TtCel45A. Interestingly, the optimal gene copy number for the chromosomal expression of TtCel45A was found to be three, indicating limitations in translational capacity, post-translational processing, and secretion, ultimately impacting protein yields in *P. pastoris*. We suggest that multiple parameters might influence a kinetic competition between protein elongation and mRNA degradation. During high-density fermentation, the highest protein concentration and endoglucanase activity of TtCel45A with three copies reached 15.8 g/L and 9640 IU/mL, respectively. At the same time, the remaining SCP of *P. pastoris* exhibited a crude protein and amino acid content of up to 59.32% and 46.98%, respectively. These findings suggested that SCP from *P. pastoris* holds great promise as a sustainable and cost-effective alternative for meeting the global protein demand, while also enabling the production of thermophilic TtCel45A in a single industrial process.

## 1. Introduction

Lignocellulose is the largest biomass resource on earth, 50% of which is cellulose [1]. Cellulose is a high-molecular-weight polymer, whose main chain is composed of Ɗ-glucose units linked via 1,4-β-glycosidic bonds [2]. As a major component of lignocellulosic biomass, cellulose utilization presents a significant opportunity for renewable energy production and sustainable development, which is a hot spot in renewable bioenergy research with enormous potential to contribute to global energy [3]. The most effective environmentally friendly and sustainable method of biomass conversion is the enzymatic digestion of cellulose using enzymes [4]. Cellulases enable the breakdown of cellulose into simpler sugars, facilitating its utilization for various applications [5]. This enzymatic conversion process holds great promise for the production of biofuels and other value-added products from cellulose, offering a sustainable alternative to dwindling petrochemical resources [6].

Cellulases are a complex group of enzymes, consisting mainly of endoglucanases (EC.3.2.1.4), exoglucanases (EC.3.2.1.91), and β-glucosidases (EC.3.2.1.21) [4], whose synergistic action is required for the complete degradation of cellulose [7]. Endoglucanases randomly cleaves β-1,4-glycosidic bonds in the polymeric structure of cellulose, exposing the reducing and non-reducing ends [8]. Finally, β-glucosidases release glucose by cleaving the disaccharide bond of cellobiose [9,10,11]. A large number of endoglucanases have been identified, primarily originating from bacteria and fungi, which have been classified into 13 glycoside hydrolase (GH) families based on their protein sequence similarity and the structure of their catalytic domain, as documented in the Carbohydrate-Active Enzyme (CAZy) database (http://www.cazy.org (accessed on 15 July 2023), [12]). Enzymes belonging to the GH5, GH7, GH12, GH44, and GH51 families catalyze reactions using a retaining mechanism, while those in the GH6, GH8, GH9, GH45, GH48, GH74, GH124, and GH131 families employ an inverting mechanism. Unlike most GH families, where enzymes with distinct substrate specificities share a similar protein fold, GH45 and GH48 members are exclusively endoglucanases. Notably, GH45 endoglucanases have demonstrated higher activity on amorphous cellulose, with great potential for various commercial applications [13].

The thermal stability of cellulase enzymes is an important parameter for many application scenarios due to the harsh conditions of biotechnological processes and enzyme storage requirements. Thermophilic cellulases, which exhibit high activity and stability at high temperatures, are particularly attractive for industrial applications [14,15,16]. While only a small number of cellulolytic enzymes from *T. terrestris* have been studied, these enzymes have demonstrated remarkable efficiency, thermal stability, and significant potential for various applications [17,18,19]. Due to its unique enzymatic properties, including remarkable thermostability, we cloned and overexpressed *T. terrestris* glycoside hydrolase family 45 (GH45) to improve the yield of enzyme for useful industrial applications. After employing multiple strategies such as increasing the copy number, site-specific integration, and high-cell-density fermentation, the heterologous thermophilic endoglucanase from *T. terrestris* produced in *P. pastoris* exhibited a high level of endoglucanase activity.

## 2. Results and Discussion

### 2.1. Overall Three-Dimensional Structure of Cel45A from T. terrestris

Cel45A from *T. terrestris* belongs to glycoside hydrolase family 45. The crystal structure of the apo-form of TtCel45A was solved at 1.42 Å, based on a crystal belonging to monoclinic unit space group P21, and one polypeptide chain in the asymmetric unit [20]. According to the relative position of the β-barrel and substrate-binding cleft, currently solved GH45 structures can be subdivided into two groups. The overall structure of TtCel45A (PDB ID: 5GLX) was very similar to that of group I GH45 endoglucanases, with a six-stranded β-barrel and a region composed of several long interconnecting loops around the substrate-binding cleft. TtCel45A consists of 389 amino acids, corresponding to a molecular weight of 43.10 kDa. The modeled structure of TtCel45a is presented in Figure 1a. The catalytic acid (Asp122) and base (Asp12) were located near the middle of the substrate-binding cleft. Another highly conserved amino acid residue, Tyr10, was located close to the active site, which was in agreement with the findings of an early study [21]. The modeled TtCel45a structure contained six disulfide bonds (Cys13-Cys136, Cys18-Cys87, Cys33-Cys57, Cys88-Cys200, Cys90-Cys190, and Cys156-Cys168; Figure 1b), which play a crucial role in thermal stability. Disulfide bridges are frequently observed in protein structures and have been extensively studied for their ability to enhance protein stability [22]. For example, the half-life of a mutant version of lipase B was increased by 4.5 times compared to that of the wild type, which was attributed to the presence of disulfide bridges [23].

### 2.2. Comparison of CRISPR-Cas9 and Homologous Recombination for Chromosomal Integration

The methylotrophic yeast *Pichia pastoris* can efficiently express heterologous genes [24,25], and is considered an ideal host for the production of recombinant proteins and chemicals [26]. Furthermore, the highly efficient assimilation of methanol, an affordable substrate, renders it a promising candidate for large-scale industrial production and a favorable platform for the biotransformation of other one-carbon compounds [27]. To improve the heterologous expression of TtCel45a, it is crucial to select an efficient genomic engineering technology. The CRISPR-Cas9 system was optimized in *P. pastoris* via the evaluation of a range of codon-optimized Cas9 genes, sgRNAs, and promoters for genomic integration and gene deletion. Non-homologous end joining (NHEJ) presents a challenge to precise genome engineering, especially for seamless gene deletion and genome integration that require homologous recombination. To compare the efficiency of CRISPR-Cas9 and homologous recombination (HR), the number of positive transformants was evaluated via flow cytometry-based cell sorting. Firstly, we constructed TtCel45a gene expression cassettes for *P. pastoris* using overlap extension PCR, incorporating a 2A peptide. The P_AOX_ promoter and T_GAP_ terminator with lengths of approximately 500 bp were ligated via homologous regions of the three fragments. In addition, we utilized a recently developed CRISPR-Cas9-meditated genome editing system [28], which entailed constructing a gRNA expression plasmid and 20 bp target sequences of gRNAs for genome targeting. The 2A peptide with high cleavage efficiency resulted in two complete gene products, extracellular TtCel5A and intracellular GFP, with a linear relationship between the enzyme expression level and fluorescence intensity. According to flow cytometric cell sorting, the CRISPR-Cas9-meditated genome editing system resulted in a 93.6% positive rate of the targeted integration of TtCel45A at PNSII-2 (Figure 2a), while HR-meditated genome editing resulted in less than a 33.33% correct integration of TtCel45A expression cassettes (Figure 2b). This low frequency suggested that NHEJ was the dominant mechanism for repairing double-strand breaks (DSBs) while HR was repressed in *P. pastoris* C1. The efficiency of this CRISPR-Cas9-mediated genome editing system was confirmed, making it suitable for use in a variety of unconventional yeasts. By optimizing the Cas9 gene and promoter through the incorporation of multiple codons for Cas9 and sgRNA expression, the CRISPR-Cas9 system was successfully optimized for the targeted cleavage of genomic DNA in *Picrosporum* sp. [29]. Additionally, a CRISPR-Cas9-mediated multigene integration method with efficient gRNA targets was developed for *P. pastoris* [30]. Therefore, CRISPR-Cas9-meditated genome editing and flow cytometry-based cell sorting were employed for further engineering, aiming at the overproduction of TtCel45A in *P. pastoris* C1.

### 2.3. Selection of Neutral Integration Sites and Optimal Copy Numbers for Enhancing Heterologous Expression of TtCel45A in P. pastoris

The availability of abundant neutral sites facilitates the genomic integration of multiple genes for more comprehensive and predictable metabolic engineering. Accordingly, the abundance of neutral loci strongly supported extensive metabolic engineering for *S. cerevisiae* [31,32], and 53 potential neutral sites were characterized in *P. pastoris*. To minimize any potential negative impact on cellular physiology and metabolism, it is crucial to carry out a determination of the heterologous expression of TtCel45A using neutral loci on the chromosome that can be modified without causing significant disruptions to the overall cellular functions [29]. These neutral loci should be identified and selected as integration sites for extensive metabolic engineering efforts [33]. The neutral sites PNSII-2, PNSII-3, PNSII-6, PNSII-8 with integration efficiencies of 100% were selected and used as the integration sites for the TtCel45a expression cassette.

While the neutral integration sites (PNSII-2, PNSII-3, PNSII-6, and PNSII-8) exhibited similar transformation efficiencies and site characteristics, a slight variation was observed in the endoglucanase activity of *P. pastoris* transformants expressing TtCel45A from each site. Under identical shake flask culture conditions, it was observed that the transformant carrying TtCel45a expression integrated at PNSII-3 displayed an endoglucanase activity of 76.07 IU/mL, whereas the transformant with integration at PNSII-6 exhibited an endoglucanase activity of 71.16 IU/mL. The influence of integration site on protein expression is affected by various factors, including the optimization of expression conditions [34], mRNA-folding effects, and global sequence features [35]. It is possible that multiple parameters modulate the kinetic competition between protein elongation and mRNA degradation. A large-scale analysis investigated the role of position effects on the expression of a GFP reporter at approximately 500 loci throughout the genome, revealing a twenty-fold difference in expression levels depending on the integration site [36]. The positioning and arrangement of genes across the genome can profoundly influence transcription, with cis-regulatory mechanisms impacting the expression of neighboring genes [37]. Studies on gene expression that do not rely on reporter constructs have provided valuable insights into the effects of position and have contributed to the hypothesis that functional gene expression clusters are influenced by genomic location [38].

To achieve the simultaneous heterologous expression of TtCel45a from two and three co-integrated expression cassettes, we conducted multi-site editing using various combinations of neutral sites. From the resulting strains, we selected three single-copy, three two-copy, and two three-copy strains to examine the correlation between gene copy number and protein expression levels. The findings revealed that integrating one to three gene copies into the chromosome enhanced TtCel45a production, but the secretion of TtCel45a was not directly proportional to the gene copy number. On average, transformants with three gene copies showed a 35.26% increase compared to those with two copies, and a 51.68% increase compared to those with a single copy (Figure 3a). Additionally, the SDS-PAGE analysis of the fermentation supernatant confirmed a gradual increase in endoglucanase activity (Figure 3b). Thus, TtCel45A was successfully produced in *P. pastoris* C1, exhibiting a single band of ~45.0 kDa. The molecular weight of TtCel45A was higher than the theoretical value due to partial glycosylation. However, strains with more than three gene copies exhibited decreased endoglucanase activity compared to what would be expected based on lower gene copy numbers. The optimal gene copy number for a high expression of TtCel45A was found to be three. However, copy numbers ranging from three to five were shown to be effective for the optimal production of various heterologous enzymes due to differences in mRNA sequence features. These results were in agreement with those of earlier reports that strains with high gene copy numbers may face limitations in translational capacity, post-translational processing, and secretion, which can impact protein yields.

### 2.4. High-Cell-Density Fermentation of P. pastoris for Cost-Effective TtCel45A Production

In recent years, the implementation of high-cell-density cultivation has become crucial for the efficient production of various microbial enzymes [39,40]. After optimizing the culture conditions in shake flask experiments, which revealed an optimal temperature of 30 °C, an inoculum volume of 5%, and an initial pH of 6.0, pilot-scale fermentation was carried out using a 5 L bioreactor. The fermentation of the TtCel45A high-expression strain with three copies was monitored at different time points via measurements of endoglucanase activity, dissolved oxygen (DO) levels, and biomass. Within 12 h, the DO value decreased significantly due to cellular growth and was maintained at approximately 20% of the atmospheric value. To promote the growth of recombinant cells, carbon sources were supplemented in the logarithmic phase of fermentation. From 12 to 120 h, the DO value remained stable, indicating that the strain utilized oxygen for both growth and enzyme synthesis. After 120 h of fermentation, the recombinant strain exhibited a significant increase in biomass, reaching a maximum OD_600_ of 420 (Figure 4a). During the methanol-fed fermentation phase, the highest protein concentration and endoglucanase activity reached15.8 g/L and 9640 IU/mL, respectively (Figure 4b). Notably, the achieved enzyme activity titer was 85.3 times higher than that in shake flasks. Therefore, high-density fermentation may be a valuable technique for achieving higher yields and the cost-effective production of heterologous endoglucanases in *P. pastoris*.

### 2.5. Analyzing the Nutritional Value of Residual Single-Cell Protein to Enhance the Economic Viability of TtCel45 Fermentation

After fermentation, the TtCel45A enzyme was mostly found in the supernatant, leaving a large amount of unused cell biomass. As a consequence, we analyzed the nutrient composition of the remaining single-cell protein (SCP). The high protein content of SCP makes it a valuable nutritional resource for both human food and animal feed. In addition to protein content, amino acid composition plays a crucial role in determining the quality of SCP [41]. The remaining SCP of *P. pastoris* exhibited a crude protein and amino acid content of up to 59.32% and 46.98%, respectively. Figure 4 illustrates the detection of 17 amino acids in total, accounting for 46.98% of the dry cell weight (DCW). Among these, 14.71% were essential amino acids, including 1.81% of isoleucine, 2.96% of leucine, 2.83% of lysine, 0.82% of methionine, 1.68% of phenylalanine, 2.06% of threonine, 2.31% of arginine, and 2.13% of valine (Figure 4c). It is worth noting that SCP from *P. pastoris* represents a rich source of the limiting amino acids lysine and methionine, which are deficient in cereals and insufficient in soy, peanuts, milk, and some meat products [42,43]. This indicates that SCP from *P. pastoris* could be mixed with other protein sources to obtain more nutritious food. Another notable finding was that the total amount of branched-chain amino acids (BCAAs), including valine, isoleucine, and leucine, accounted for 6.90% of the 17 detected amino acids, representing 14.69% of the total. BCAAs serve as important building blocks in the body, promoting glucose uptake and increasing ATP production [44]. Furthermore, BCAAs play a role in regulating body lipid metabolism, protein synthesis, and the immune response [41,45]. Thus, SCP from *P. pastoris* is also a source of BCAAs that can be added to low-protein foods. In summary, SCP from *P. pastoris* (59.32% in this report) generally contained a higher percentage of protein than it did that of soy (38.6%), fish (17.8%), meat (21.2%), and whole milk (3.28%) [41].

Compared to traditional protein sources, *P. pastoris* offers several advantages in terms of production speed and efficiency. Unlike other sources that require longer cultivation periods, *P. pastoris* enables the rapid accumulation of biomass and the subsequent production of SCP [46]. This time-saving aspect enhances overall production efficiency and streamlines the manufacturing process. Additionally, SCP production from *P. pastoris* requires significantly less land than traditional protein sources do [47]. The ability to achieve high biomass yields in a smaller cultivation area is particularly advantageous in resource-limited environments. This not only optimizes space utilization, but also minimizes the environmental impact associated with protein production. The overproduction of SCP by engineered *P. pastoris* offers potential solutions for achieving a low-carbon future and addressing food shortages [47]. By synthesizing substitute proteins, engineered *P. pastoris* represents a cost-competitive approach for the industrial bio-manufacturing of artificial food and feed, contributing to global food security and sustainable agriculture [48].

Furthermore, the production of SCP from *P. pastoris* is not constrained by weather conditions [49,50]. While traditional protein sources heavily rely on suitable climatic conditions for cultivation, *P. pastoris* can be cultivated in a controlled bioreactor environment, ensuring consistent and reliable production outcomes throughout the year [51]. This independence from weather fluctuations guarantees a stable and uninterrupted supply of SCP, regardless of seasonal variations or adverse weather events. In summary, SCP production from *P. pastoris* offers superior production efficiency compared to traditional protein sources, owing to its shorter production time, reduced land requirements, and weather independence. These advantages make SCP from *P. pastoris* a promising and sustainable alternative for addressing the rising global demand for protein while achieving the cost-effective production of the thermophilic endoglucanase TtCel45A at the same time (Figure 4d).

## 3. Materials and Methods

### 3.1. Strains and Culture Conditions

*P. pastoris* C1 was obtained via ARTP mutagenesis in our laboratory and preserved in the China General Microbiological Culture Collection Center (CGMCC no. 24324). Unless otherwise specified, *P. pastoris* C1 and its derivatives were cultivated in YPD medium containing 20 g/L of glucose, 20 g/L of peptone and 10 g/L of yeast extract. To select transformants, YPD medium was supplemented with 100 mg/L of geneticin (G418). *E. coli* DH5 was grown in LB medium (10 g/L of tryptone, 10 g/L of NaCl, and 5 g/L of yeast extract) and kanamycin (100 mg/L) was added to promote plasmid retention. All yeast strains were cultured in a shaker at 30 °C and 200 rpm.

### 3.2. Plasmid and Strain Construction

The coding sequence spanning the catalytic domain (residues 22–299) of TtCel45A (GenBank access no. XP 003651003.1) was chemically synthesized. The genetic manipulation of *P. pastoris* was based on the CRISPR-Cas9-meditated genome editing system, as previously described [28]. To construct gRNA expression plasmids, 20 bp target sequences of gRNAs for genome targeting were designed using a user-friendly online web tool (CRISPR RGEN Tools, http://www.rgenome.net/, accessed on 1 May 2022). The plasmid pPICZ-Cas9-gAOX1 and donor DNA with specific mutations were used to co-transform competent cells of *P. pastoris*, and the transformants were grown for three days on YPD agar plates containing 100 mg/L of Zeocin. The mutants were verified via gene sequencing. Single colonies were picked from the plates and used to inoculate 5 mL of YPD medium, after which the resulting seed culture was used to inoculate 50 mL of BMGY (1% yeast extract, 2% peptone, 100 mM potassium phosphate at pH 6.0, 1.34% yeast nitrogen base with ammonium sulfate without amino acids, 4 × 10^−5^% biotin and 1% glycerol) at 30 °C for 24 h. Cells were harvested via centrifugation and resuspended in 50 mL of BMMY (1% yeast extract, 2% peptone, 100 mM potassium phosphate at pH 6.0, 1.34% yeast nitrogen base with ammonium sulfate without amino acids, 4 × 10^−5^% biotin, and 0.5% methanol) and incubated at 30 °C for 24 h. Subsequently, 1% methanol was supplemented every 24 h to sustain expression from the AOX1 promoter for 4–5 days.

### 3.3. Modeling of Endoglucanase Cel45A from Thermothielavioides terrestris

The three-dimensional structure of endoglucanase Cel45A from *Thermothielavioides terrestris* was modeled using the crystal structure of TtCel45A from *T. terrestris* (PDB ID: 5GLX, [52]).

### 3.4. Enzyme Activity Measurements

Endoglucanase activity was determined using the dinitrosalicylic acid (DNS) method as previous described [53], with minor modifications as follows. In this study, equal amounts of enzyme solution (50 mM sodium citrate buffer, pH 4.5) and 1% (*w*/*v*) carboxy-methyl cellulose (CMC) were mixed and incubated in a water bath at 50 °C for 10 min. The reaction was then mixed with DNS and incubated in boiling water for 5 min to destroy residual enzyme activity. After cooling in a cold water bath for 5 min, the absorbance at 540 nm was measured to calculate enzyme activity. One unit of endoglucanase was defined as the amount of enzyme that releases 1 μmol of product per minute under the described assay conditions.

### 3.5. Fed-Batch Fermentation

The fermentation was conducted in a 5 L bioreactor (Baoxing; Shanghai; China) with a working volume of 3 L. The initial seed culture was grown in YPD medium at 30 °C until the OD_600_ reached 10, at which point 150 mL was transferred into the fermenter. During the exponential growth phase, the fermenter maintained an ideal pH (4.8) and temperature (30 °C). The first step of feed addition was initiated when the nutrients in the fermenter were depleted, as evidenced by a rebound in the dissolved oxygen (DO) and pH values. The feeding medium containing 30% (*w*/*v*) glycerol, 1.2% PTM1, and 0.12% biotin was depleted at around 12 h. The DO level was kept at approximately 20% of the atmospheric value. Samples were collected every 3 h to measure the residual carbon source, OD_600_, and biomass. When the glycerol was depleted and the dissolved DO level exceeded 70%, the flow plus methanol-induced feeding phase was initiated. During the injection of methanol, the temperature was kept at 28 °C and the dissolved oxygen concentration was maintained at 25%. Samples were taken every 24 h to determine the enzyme activity, biomass, and nitrogen content of the fermentation broth.

### 3.6. Biomass and Crude Protein Analysis

The biomass was characterized via the dry cell weight (DCW, g L^−1^) and optical density at 600 nm (OD_600_). A 10 mL sample was collected every 6 h and centrifuged at 4200× *g* at 4 °C for 10 min. The supernatant was discarded, after which the cells were washed three times and resuspended in 10 mL of 100 mM Na phosphate buffer (pH 7.0). The tubes containing centrifuged cells were dried at 105 °C to a constant weight to determine the DCW. The relationship between the OD_600_ and DCW was represented via empirical Equation (1), OD_600_ L^−1^ = 0.24 g_DCW_ L^−1^ [54]. The nitrogen content of the homogenized residues was analyzed using the Kjeldahl method [55]. A conversion factor of 6.25 was used to convert the nitrogen content into the protein content.

## Figures and Tables

**Figure 1 ijms-24-15017-f001:**
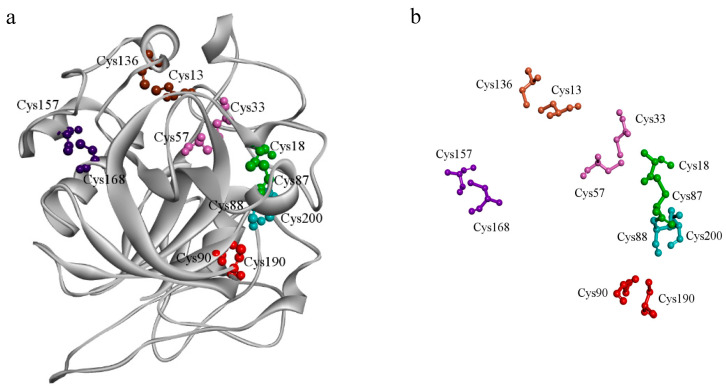
Overall structure of TtCel45A from *Thermothielavioides terrestris.* (**a**) The three-dimensional structure of Cel45A from *Thermothielavioides terrestris.* (**b**) Six disulfide bonds in the homology model of TtCel45a.

**Figure 2 ijms-24-15017-f002:**
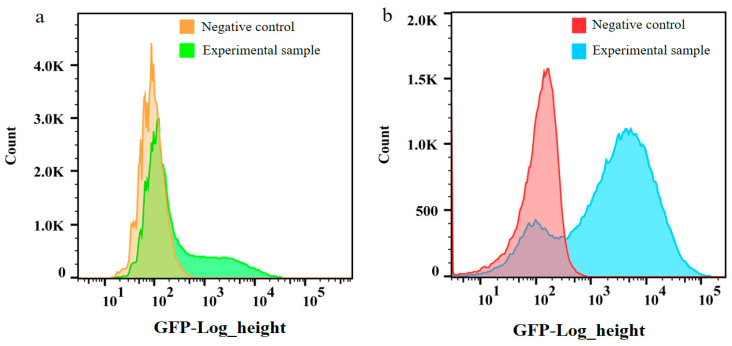
Percent of cells successfully transformed with the GFP expression cassette using (**a**) CRISPR-Cas9 and (**b**) homologous recombination, as detected via flow cytometry.

**Figure 3 ijms-24-15017-f003:**
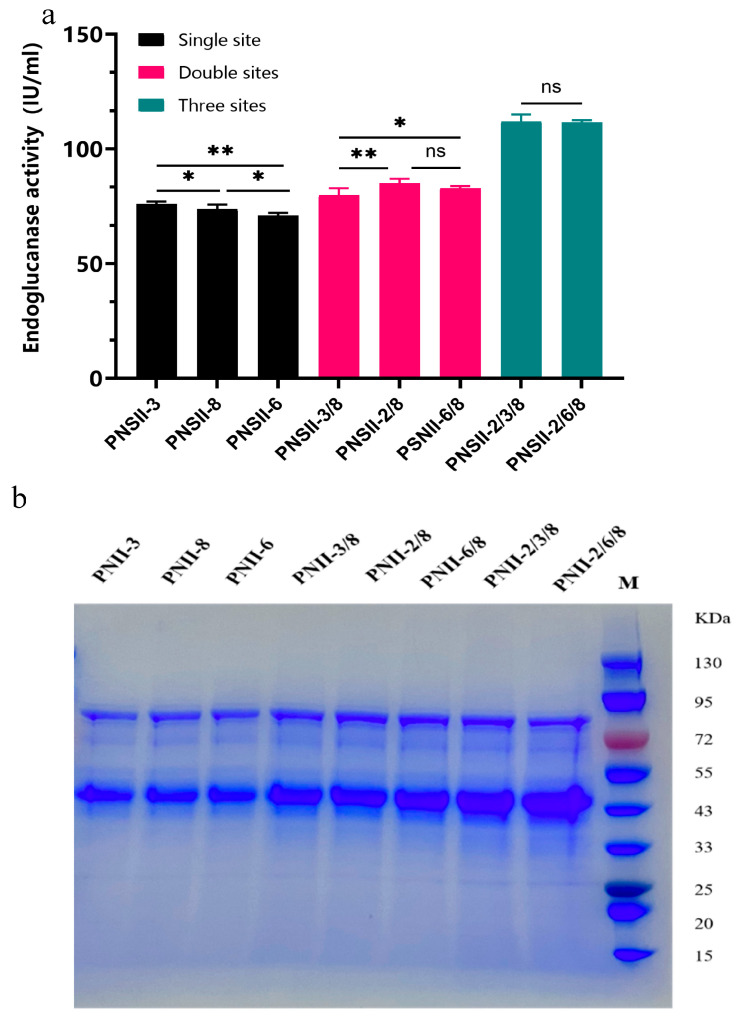
Comparison of TtCel45A expression in transformants with different copy numbers of the genomically integrated expression cassette. (**a**) Comparison of endoglucanase activity of transformants with different copy numbers. “*” for *p* < 0.05, “**” for *p* < 0.001, and “ns” for no significance. (**b**) SDS-PAGE of TtCel45A expression in transformants with different copy numbers. Red marker for 72KDa.

**Figure 4 ijms-24-15017-f004:**
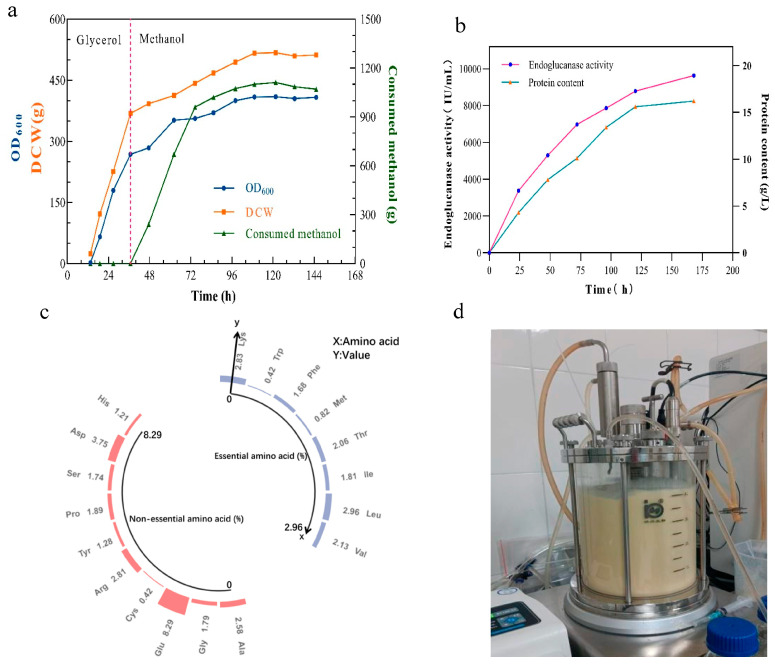
Analysis of fed-batch fermentation and nutritional content of *P. pastoris* biomass after fermentation. (**a**) Fed-batch fermentation of *P. pastoris* in a 5 L bioreactor. (**b**) Analysis of endoglucanase activity during fermentation. (**c**) Amino acid profile of single-cell protein from *P. pastoris* co-produced in fermentation. (**d**) High-cell-density fermentation of TtCel45A in a 5 L bioreactor.

## Data Availability

All data is included in the manuscript.

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
