# Peer review of "Enhancing the Heterologous Expression of a Thermophilic Endoglucanase and Its Cost-Effective Production in Pichia pastoris Using Multiple Strategies"

_ijms, 2023, doi:10.3390/ijms241915017_

Round 1
Reviewer 1 Report
The reviewed manuscript, titled "Enhancing the heterologous expression of a thermophilic endoglucanase and its cost-effective production in Pichia pastoris using multiple strategies” by Dai et al. is a solid work that characterizes the effects of synthetic integration of an endoglucanase for biofuel production. Overall, this work is sound, but could benefit from revision to the work as outlined below. Furthermore, this work does not meet the threshold for being characterized as a full Article. It should be classified as a Communication or another comparable short form publication to accurately reflect the work done.
Major comments:
Figure 1: Since the authors did not solve the crystal structure of TtCel45A in this work, please provide accession numbers for where the structure/data was obtained (e.g. the PDB or comparable repository). Part b does not seem useful as depicted, as the six disulfide bonds do not appear to related to the structure in part A. For example, Cys90 and Cys190 would appear to be floating in space relative to the ribbon diagram in part A. Please revise this figure, I recommend highlighting the sites of the specified disulfide bridges on the actual structural image for clarity for the readers.
Line 182: this paragraph could use a clearer transition. State what the cloning goal/overarching aim of the comparison of the two gene editing approaches is prior to simply stating you compared CRISPR to HR-mediated chromosomal modification.
Line 194: “The CRISPR-Cas9 system revolutionized genome editing due to its efficiency, accuracy, and convenience”. Im not sure that this is relevant to the authors results. This is obvious and has been demonstrated repeatedly – as the authors point out in their Cai et al., 2019 reference.
Figure 2: Changing the Y-axis on both graphs so that they are the same would better highlight the efficiency of the two approaches. More concerningly, this figure simply seems to show GFP fluorescence. I am unable to relate what is shown in this figure to the statements on line 191-192 regarding the correct integration of the cassette from the GFP counts. If anything this figure seems to show that the HR-approach generated more fluorescence, meaning that it was more effective in the integration of the reporter construct used based on the simple fluorescence provided. While I don’t doubt that CRISPR is effective, the authors have not shown evidence to support their statements in this section (it appears to be the section from 191-202). If this is important in the authors opinion, they should sequence and/or use targeted PCR to verify the integration of this reporter at the targeted genomic site. It is this reviewers opinion that this is ultimately irrelevant to the rest of the manuscript – this can simply be removed entirely to improve the work accordingly.
Line 222: I suggest selecting one or more references that are more recent than a decade old. Here is a suggestion (disclaimer – selection of references is entirely up to the authors and will not affect this reviewer’s comments or decisions about this work).
Arnone, J.T., 2020. “Genomic Considerations for the Modification of Saccharomyces cerevisiae for Biofuel and Metabolite Biosynthesis” Microorganisms.
PMID: 32110897 PMCID: PMC7143498 DOI: 10.3390/microorganisms8030321
In general, the authors rely very heavily on relatively few references. Adding more would strengthen this work significantly.
Line 253: Please provide citations and references for the earlier reports that your data is in agreement with for comparison.
Line 297: The authors do not refer to figure 3b in this work. Please address this or remove this portion of the figure. If the authors choose to discuss this, please provide context to allow the reader to understand what is being shown. An undescribed SDS-PAGE gel in its own provides little support of anything and has no use otherwise.
Integration effects and expression has been discussed and studied in a number of works. This reviewer recommends adding the following to their discussions and elaborating on the significance of ‘neutral sites’ – or what some researchers term ‘safe harbours’ for modification and integration. Many regions of the genome interact extensively at a longer distance than might be otherwise expected – resulting in the clustering of functionally related genes over evolutionary time and increases the chances of secondary effects at a distance.
This is a rather important and excellent component of the manuscript and could benefit from additional context and discussion by the authors. This reviewer provides the following suggestions (disclaimer – selection of references is entirely up to the authors and will not affect this reviewer’s comments or decisions about this work).
Genomic clustering within functionally related gene families in Ascomycota fungi
D Hagee, AA Hardan, J Botero, JT Arnone
Computational and Structural Biotechnology Journal 18, 3267-3277
Functional Clustering of Metabolically Related Genes Is Conserved across Dikarya
GM Cittadino, J Andrews, H Purewal, P Estanislao Acuña Avila, ...
Journal of Fungi 9 (5), 523
Enhancer Sharing Promotes Neighborhoods of Transcriptional Regulation Across Eukaryotes
Porfirio Quintero-Cadena and Paul W. Sternberg
G3 (Bethesda). 2016 Dec; 6(12): 4167–4174.
Minor comments
Line 82: In every other subheading, the first letter of each word is only capitalized at the start of the sentence, while here that is the case for every letter. Please revise for consistency.
Language is fine
Reviewer 2 Report
Dear Sirs, In my opinion manuscript need only minor improvements in the text before publication. Please see the details in the file attached. My main request is to express DO concentration in g/L not as % - becasue it was hard to guess the % of what it means - concentration in air or % of maximal saturation for specific medium.- thus i think g/L seems more universal

Round 2
Reviewer 1 Report
I am satisfied with the revisions made and the justifications provided by the authors. I support publication in the current form.